# Riding the Omicron BA.5 Wave: Improved Humoral Response after Vaccination with Bivalent Omicron BA.4-5-Adapted mRNA SARS-CoV-2 Vaccine in Chronic Hemodialysis Patients

**DOI:** 10.3390/vaccines11091428

**Published:** 2023-08-28

**Authors:** Eugen Ovcar, Sammy Patyna, Niko Kohmer, Elisabeth Heckel-Kratz, Sandra Ciesek, Holger F. Rabenau, Ingeborg A. Hauser, Kirsten de Groot

**Affiliations:** 1Department of Internal Medicine 4, Nephrology, University Hospital, Goethe University Frankfurt, 60590 Frankfurt am Main, Germany; eugen.ovcar@kfh.de (E.O.); patyna@em.uni-frankfurt.de (S.P.);; 2KfH Kuratorium for Dialysis and Transplantation, 63069 Offenbach am Main, Germany; 3Department of Internal Medicine III, Internal Medicine, Nephrology, Rheumatology, Sana Klinikum, 63069 Offenbach am Main, Germany; 4Institute for Medical Virology, University Hospital, Goethe University Frankfurt, 60596 Frankfurt am Main, Germany; niko.kohmer@kgu.de (N.K.); sandra.ciesek@kgu.de (S.C.); rabenau@em.uni-frankfurt.de (H.F.R.); 5KfH Kuratorium for Dialysis and Transplantation, 64521 Gross-Gerau, Germany; elisabeth.heckel-kratz@kfh.de; 6German Center for Infection Research, External Partner Site, 60596 Frankfurt am Main, Germany; 7Fraunhofer Institute for Molecular Biology and Applied Ecology (IME), Branch Translational Medicine and Pharmacology, 60596 Frankfurt am Main, Germany; 8University Hospital, Goethe University Frankfurt, 60590 Frankfurt am Main, Germany

**Keywords:** dialysis, SARS-CoV-2, Omicron, vaccination, humoral response

## Abstract

Hemodialysis patients faced an excess morbidity and mortality during the COVID-19 pandemic. We evaluated the effect of second-generation mRNA vaccines against Omicron BA.4 and BA.5 variants of SARS-CoV-2 on humoral immunity. The study population comprised 66 adult hemodialysis patients who have encountered four SARS-CoV-2 antigen contacts through vaccination or infection. We assessed their humoral response using an anti-SARS-CoV-2 spike receptor binding domain IgG antibody assay (S-RBD-ab), measuring neutralizing antibodies against ancestral strain of SARS-CoV-2, Delta, and Omicron in a surrogate virus neutralization test (SVNT), and specifically against BA.5 in a plaque reduction neutralization test (PRNT) before and four weeks after vaccination with Comirnaty Original/Omicron BA.4-5. During the following six months, SARS-CoV-2 infections and symptom severity were documented. The bivalent mRNA vaccine led to a 7.6-fold increase in S-RBD-ab levels and an augmented inhibition of the Omicron variant in SVNT by 35% (median). Seroconversion in the Omicron BA.5-specific PRNT was attained by in 78.4% of previously negative patients (29/37). Levels of S-RBD-ab correlated with inhibition in the Omicron-specific SVNT and neutralization titers in the BA.5-PRNT. Eleven SARS-CoV-2 infections occurred in the six-month follow-up, none of which took a life-threatening course. The bivalent mRNA vaccine improved the SARS-CoV-2 virus variant-specific humoral immunity in chronic hemodialysis patients. Measurement of S-RBD-ab can be used in hemodialysis patients to estimate their humoral immunity status against Omicron BA.5.

## 1. Introduction

Patients on maintenance dialysis show humoral and cellular immune deficiency [1,2,3], which renders them more prone to infections. It has been previously reported that dialysis patients have a reduced immune response to influenza or hepatitis B virus vaccines [4,5]. 

In the pre-vaccination era of the SARS-CoV-2 pandemic, it has been shown that COVID-19 disease claimed more victims among patients on dialysis compared to the general population [6]. Continuous evolution of SARS-CoV-2 virus and progressive immunization of the population as a result of vaccination and/or infections changed the situation: the COVID-19 mortality of dialysis patients decreased from 20–30% to 2.2% [7]. However, new challenges include a higher transmission rate of emerging virus variants and a reduced protection after vaccination with the wild-type mRNA vaccine [8,9]. The second-generation mRNA vaccines were designed to be bivalent: 50% of the anti-wild-type strain and the adapted vaccine (firstly to the Omicron BA.1 variant and later to BA.4 and BA.5), each. In immunocompetent patients, this led to an improved humoral vaccination response [10,11]. Its effect in hemodialysis patients was not evaluated in the approval studies and first publications.

In autumn 2022, strict containment strategies were reduced in Germany. Subsequently, a steep rise in incidence of SARS-CoV-2 infections was observed [12]. The majority of our hemodialysis patients had received their last wild-type SARS-CoV-2 vaccination or became infected with this virus between February and April 2022. The Robert Koch Institute (the German government research institute responsible for infection disease control) recommended a booster with the bivalent mRNA vaccine for immunodeficient patients who already had four antigen contacts to the spike protein of SARS-CoV-2 (by either infection or vaccination), at least six months after the last one. This study was designed to explore the effect of a bivalent mRNA vaccine on the humoral immune response and clinical outcomes in chronic maintenance hemodialysis patients.

## 2. Materials and Methods

### 2.1. Study Design and Population

We investigated adult hemodialysis patients in tertiary care dialysis units at the outpatient KfH Dialysis Centers in Offenbach, Gross-Gerau, and at the Department of Nephrology and Dialysis of Sana Hospital Offenbach, Germany, in a prospective, investigator-initiated, non-interventional, observational trial. 

Out of 207 patients treated in the three mentioned dialysis centers, we report on a subgroup of 66 patients who have had four exposures to the spike protein of SARS-CoV-2 by either infection or vaccination (Figure 1, patient recruitment). All enrolled patients were vaccinated with the adapted COVID-19 vaccine Comirnaty Original/Omicron BA.4-5 in October 2022, hereafter also referred to as fifth antigen contact. 

We assessed the humoral response to the vaccination before (t_0_) and four weeks after the vaccination (t_1_) with the bivalent mRNA vaccine (Figure 2).

We measured IgG antibody levels targeting the SARS-CoV-2 spike receptor binding domain (S-RBD-ab), potentially neutralizing antibodies by surrogate virus neutralization test (SVNT) against wild-type (Wuhan Hu-1), Delta (B.1.617.2), Omicron (B.1.1.529), and neutralizing antibody titers specific for BA.5 measured with a plaque reduction neutralization test (PRNT).

During the six months following this vaccination, SARS-CoV-2 infections and symptom severity according to the World Health Organization guidelines were documented [13]. 

In all centers, filtering face piece (FFP2) masks were still compulsory for patients and staff.

### 2.2. Methods

#### 2.2.1. Assessment of Anti-SARS-CoV-2 Spike-Specific Antibodies

Antibodies targeted against the SARS-CoV-2 spike receptor binding domain (S-RBD-ab) were measured using the Abbott SARS-CoV-2 IgG II Quant and SARS-CoV-2 IgG (Abbott GmbH, Wiesbaden, Germany) on the Abbott Alinity i platform. The cut-off for positivity was 7.1 BAU/mL (manufacturer’s specification). To assess immunological protection against severe SARS-CoV-2 infection, we used the threshold for protection as 500 BAU/mL of the Delta variant [14].

#### 2.2.2. Assessment of SARS-CoV-2-Specific Surrogate Virus Neutralization Test against Ancestral Strain (Wuhan Hu-1), Delta (B.1.617.2), and Omicron (B.1.1.529)

The inhibition of S-RBD binding to angiotensin-converting enzyme 2 (ACE2) for the detection of SARS-CoV-2-neutralizing antibodies was assessed with an ELISA-based GenScript SARS-CoV-2 Surrogate Virus Neutralization Test Kit (GenScript Biotech, PiscataWay Township, NJ, USA). The threshold for positivity was defined as ≥50% of inhibition for the ancestral strain (Wuhan Hu-1) and Delta (B.1.617.2) and ≥70% for the Omicron (B.1.1.529) SVNT.

#### 2.2.3. Assessment of SARS-CoV-2 Omicron BA.5-Specific Neutralizing Antibodies with a Plaque Reduction Neutralization Test (PRNT) 

Inactivated patients’ serum samples were diluted 1:10 in cell culture medium and thereafter serially diluted (1:2) and incubated with 4000 TCID50/mL (median (50) tissue culture infectious dose per milliliter) of the Omicron variant BA.5 of SARS-CoV-2 for one hour. 

CaCo-2 (human colon carcinoma) cells were cultured in Minimum Essential Medium (MEM) supplemented with 10% fetal calf serum (FCS), 100 IU/mL of penicillin, and 100 g/mL of streptomycin. All culture reagents were purchased from Sigma (St. Louis, MO, USA). The CaCo-2 cells were originally obtained from DSMZ (Braunschweig, Germany, no.: ACC 169), differentiated by serial passaging and selected for high permissiveness to virus infection [15].

Cultured Cancer Coli (CaCo)-2 cells were subsequently inoculated for 72 h with the prepared patients’ sera/virus mix. Thereafter, CaCo-2 cells were analyzed for cytopathic effect (CPE) formation using light microscopy to define the neutralization titers as reciprocal value of the highest dilution of serum that prevented infection of 50% of the cells. The criteria for CPE are described in detail before [16]. Each serum sample was tested in duplicate, in the case of discrepancies, and the lowest observed titer was chosen. Cut-off for positivity was set to ≥1:20 dilution.

#### 2.2.4. Detection of SARS-CoV-2 Infections

Polymerase chain reaction (PCR) testing of nose and throat swabs was performed in case of symptoms suggesting respiratory infection. Only a fraction of the patients (Sana Hospital Offenbach, *n* = 11) was tested weekly using SARS-CoV-2 PCR. 

The SARS-CoV-2 PCR test was performed as dual target PCR for ORF1 and E genes on a Cobas^®^ 6800/8800 system (ROCHE Diagnostics, Mannheim, Germany).

### 2.3. Study End Points

The primary end point of this study was the percentage of patients with a positive seroconversion in PRNT specific for Omicron BA.5 four weeks after the vaccination with the bivalent mRNA vaccine in the patient subgroup with negative PRNT prior to this booster. 

Secondary end points were (a) the SARS-CoV-2 infection rate after the vaccination and (b) the occurrence of life-threatening COVID-19 disease and COVID-19-associated mortality despite five past exposures to SARS-CoV-2 during the six-month follow-up.

### 2.4. Statistics 

Continuous variables were presented as medians and interquartile ranges (IQR) and categorical variables were reported as frequencies and percentages. To differentiate between patient cohorts, the Fisher’s exact test for categorical variables was used. For quantitative variables, a t-test or a Mann–Whitney-U test was applied for parametric and nonparametric data, respectively. For comparison of more than two groups, one-way analysis of variance (ANOVA) followed by Bonferroni’s post hoc correction or the Kruskal–Wallis test followed by Dunn’s post hoc test was performed for parametric and nonparametric data, respectively. For related samples comprising more than two groups, Friedman test was performed. With means of the Spearman’s rank correlation analysis, bivariate relationships by calculating Spearman’s Rho (correlation coefficient) were determined. Normality was assessed using the Kolmogorov–Smirnov test. To evaluate the determinants of independent predictors of response rate to vaccination, univariate logistic regression analyses were performed and odds ratios (OR) and corresponding 95% confidence intervals were calculated.

All *p*-values reported were two-sided, and the level of significance was set to *p* < 0.05. Statistical analyses were performed using GraphPad Prism version 9.3.1 (GraphPad Software, San Diego, CA, USA) and BiAS (v11.01; epsilon-Verlag, Nordhastedt, Germany).

## 3. Results

### 3.1. Basic Study Cohort Characteristics

The demographic and clinical characteristics of the 66 enrolled patients are described in Table 1. The applied vaccines and infection status are shown in Table 2. Prior to the bivalent vaccine, 13 patients had experienced 14 SARS-CoV-2 infections. The SARS-CoV-2 Omicron BA.5 variant was dominant in Germany during the first half of this trial, followed by the XBB.1 sublines [17].

### 3.2. Humoral Immunity before the Vaccination (Time Point t_0_)

BA.5-specific PRNT positivity was detected in 43.9% (29/66) of our patients before the vaccination, defined as a titer ≥ 1:20. Nineteen of these 29 patients (65.5%) had not previously been infected with SARS-CoV-2. Thus, they were able to neutralize Omicron BA.5 in PRNT after vaccination with monovalent BNT162b2 (Comirnaty) and/or mRNA-1273 (Moderna) vaccines developed against the wild-type virus, having had the last booster eight months ago.

However, the univariate logistic regression analysis revealed that patients who experienced a SARS-CoV-2 infection among their four previous antigen contacts had a twelve times higher chance for a positive result in the Omicron-SVNT and a four times higher chance for positivity in Omicron BA.5-specific PRNT compared to those patients without such an infection (odds ratios, OR 12.4, *p* = 0.003; OR 4.35, *p* = 0.028, respectively). 

### 3.3. Humoral Response Four Weeks after the Vaccination (Time Point t_1_)

The primary end point of this trial, a positive seroconversion in the PRNT specific for Omicron BA.5, was attained by 78.4% (29/37) of the patient subgroup that was negative in the Omicron BA.5-PRNT before the vaccination. 

Four weeks after the vaccination, 87.9% of the patients (58/66) showed a positive result in BA.5-specific PRNT (Figure 3A), which equals a two-fold increase in response rate compared with the time point t_0_ before the booster vaccine (t_0_: 43.9% (29/66) vs. t_1_: 87.9% (58/66), *p* < 0.0001). Notably, this fifth antigen contact induced a distinct increase, especially in neutralization titers of 1:160 and above (t_0_: 10.6% (7/66) vs. t_1_: 51.5% (34/66), Figure 3B), indicating a much higher level of humoral immune response. 

Humoral immunity and clinical parameters of the Omicron BA.5-PRNT positive and negative patients did not reveal significant differences after the vaccination (Table 3). Only the drug-induced immunosuppression was associated with a negative PRNT (*p* = 0.0012), whereby the small size of this cohort (eight non-responders) limits the informative value.

The vaccination with the Omicron BA.4-5-adapted mRNA vaccine resulted in a 7.6-fold increase in SARS-CoV-2-specific spike receptor binding domain antibody levels (S-RBD-ab, *p* ≤ 0.001, Figure 4A) and an elevation of 30% in response rate (S-RBD-ab-levels > 500 BAU/mL, *p* ≤ 0.001). 

The surrogate virus neutralization test (SVNT) showed a significant increase in inhibition against the Omicron variant after the vaccination. The response rate was enhanced by 40.9% (inhibition in Omicron-SVNT ≥ 70%, t_0_: 43.9% (29/66) vs. t_1_: 84.8% (55/66), *p* ≤ 0.001), and the median inhibition rate of potentially neutralizing antibodies (median %-INH) increased by 35% (t_0_: median (IQR) 60%-INH (0–94.3) vs. t_1_: 95%-INH (86.8–98) (*p* < 0.001), Figure 4D). 

Inhibition in the SVNT against wild-type SARS-CoV-2 (WT, Figure 4B) and Delta variant (Figure 4C) was already high prior to the bivalent vaccination. The response rate (inhibition ≥ 50%) increased from 93.9% (t_0_: 62/66) to 97.0% (t_1_: 64/66) in WT-SVNT and from 86.4% (t_0_: 57/66) to 97.0% (t_1_: 64/66) in Delta-SVNT, respectively. The median values were not improved by the BA.4-5 vaccine (WT t_0_: median (IQR) 97%-INH (95–97) vs. t_1_: 95%-INH (94–95) (*p* < 0.0001); Delta t_0_: median (IQR) 97%-INH (88.8–98) vs. t_1_: 96%-INH (95–96) (*p* = 0.0007)). 

Furthermore, our data indicate that levels of S-RBD-ab correlated significantly with the inhibition in Omicron-SVNT (Table 4; t_0_: rho = 0.81, *p* < 0.0001; t_1_: rho = 0.68, *p* < 0.0001) and neutralization titers in BA.5-PRNT (Table 4, t_0_: rho = 0.63, *p* < 0.0001; t_1_: rho = 0.75, *p* < 0.0001). 

There was also a strong correlation between the neutralization titers for Omicron variant of SARS-CoV-2 in SVNT and Omicron BA.5-PRNT (Table 5; t_0_: rho = 0.68, *p* < 0.001; t_1_: rho = 0.72, *p* < 0.001).

### 3.4. Clinical Outcomes

The bivalent vaccine did not cause any moderate or severe adverse events (Grade 2–4 on the FDA toxicity grading scale). Mild reactions to the vaccine were not recorded.

In the first four weeks after the vaccination with bivalent mRNA vaccine, no symptomatic infections with SARS-CoV-2 occurred among the study participants despite high infection rates in the general population and reduction of the contact restrictions in public life (Figure A1). However, subsequently eleven SARS-CoV-2 infections occurred between four weeks and six months following the vaccination (secondary end point a). Ten patients showed mild symptoms and one patient suffered from moderate COVID-19 disease (pneumonia). This patient had a low neutralization titer in the PRNT (1:20). 

Life-threatening COVID-19 disease, despite five past exposures to SARS-CoV-2 (secondary end point b), did not occur during the six-month follow-up.

## 4. Discussion

Studies on the first and second booster vaccinations against SARS-CoV-2 demonstrated how the S-RBD-ab level decreased rapidly after two to six months after vaccination [18,19,20,21]. 

Congruently, our findings show low S-RBD-ab level at the time point t_0_ before the vaccination with the bivalent vaccine (six to eight months after the fourth SARS-CoV-2 antigen exposure). Nevertheless, the activity of the neutralizing antibodies was sufficient to prevent cytopathic effect in the Omicron BA.5-PRNT in 43.9% (29/66) of patients. One-third (34.5%) of those has had a previous breakthrough infection. Our data suggest that protection is more robust with the combination of repeated vaccinations and SARS-CoV-2 infection than with the same number of vaccinations alone. Consistent with our findings, Huth et al. [22] also reported significantly higher anti-spike IgG concentrations and neutralization activity in hemodialysis patients with breakthrough infections as compared to vaccination only.

The bivalent mRNA-vaccine against COVID-19 (Comirnaty Original/Omicron BA.4-5) led to an increase in S-RBD-ab levels and neutralizing capacity against Omicron, the prevalent SARS-CoV-2 subtype at the time of the study. The humoral immunity (assessed with neutralization tests) against wild-type SARS-CoV-2 and Delta variants was already present before the vaccination. The intention for the choice of the bivalent Omicron BA.4/BA.5-adapted vaccines in the general population was to enhance neutralization breadth and to confer protection to individuals with no preexisting immunity against SARS-CoV-2 [23]. In view of our data, probably the anti-wild-type strain vaccine component would not have been mandatory for our hemodialysis patients exposed to SARS-CoV-2 antigen (mostly to the wild-type) four times within 18 months. 

A good correlation of the S-RBD-ab levels with a wild-type SARS-CoV-2 PRNT has already been reported [24,25]. Espi et al. [26] established that S-RBD-ab ≥ 997 BAU/mL were systematically associated with positivity in PRNT against WT of SARS-CoV-2 in patients receiving maintenance hemodialysis. 

Huth et al. [22] showed that the S-RBD-ab concentration after booster vaccination was the most important predictor of high neutralization activity against BA.4 and BA.5. Our data confirm that levels of S-RBD-ab correlate significantly with inhibition in the Omicron-specific SVNT and the titer of the Omicron BA.5-PRNT both before and four weeks after the vaccination. In our cohort, S-RBD-ab titer ≥ 2240.4 BAU/mL after the vaccination with the bivalent Omicron BA.4-5-adapted mRNA vaccine were associated with positivity in the Omicron-SVNT and Omicron-BA.5-PRNT.

The limitation of our study is the rather small size of the cohort (66 participants, eleven SARS-CoV-2 infections in the study period). The homogeneity and stringent characterization of the studied patients (exactly five antigen contacts to the SARS-CoV-2 spike protein) represents a strength of the study.

Due to the progressive immunization of the population against SARS-CoV-2 through vaccination and infections, the COVID-19 pandemic is moving on to an endemic phase. Mortality from the disease induced by the currently circulating strains has substantially decreased; however, contagiousness has risen continuously. Thus, the immune-deficient dialysis patients’ population remains endangered. It is also important to reduce SARS-CoV-2 transmission within the dialysis facilities to avoid supply shortages caused by simultaneous disease outbreaks among patients and staff. As a result of the high vaccination rate, there were no COVID-19 outbreaks in our dialysis centers during the autumn wave of 2022.

## 5. Conclusions

The bivalent vaccination improved the humoral IgG-related BA.4-5-specific immune response in more than half of the investigated hemodialysis patients. This translated in a low COVID-19 disease rate of mostly mild courses from four weeks after the bivalent vaccine onwards.

As there was a good correlation between variant-specific plaque reduction neutralization test results, ELISA-based variant-specific surrogate virus neutralization test, and SARS-CoV-2 wild-type antibody response as measured in immunoassay, one of the last two mentioned may be sufficient for clinical practice to estimate the patient’s humoral immunity status even against the BA.5 variant.

Annually adapted SARS-CoV-2 vaccinations could be an instrument together with seasonal influenza vaccination to protect the vulnerable group of hemodialysis patients. 

## Figures and Tables

**Figure 1 vaccines-11-01428-f001:**
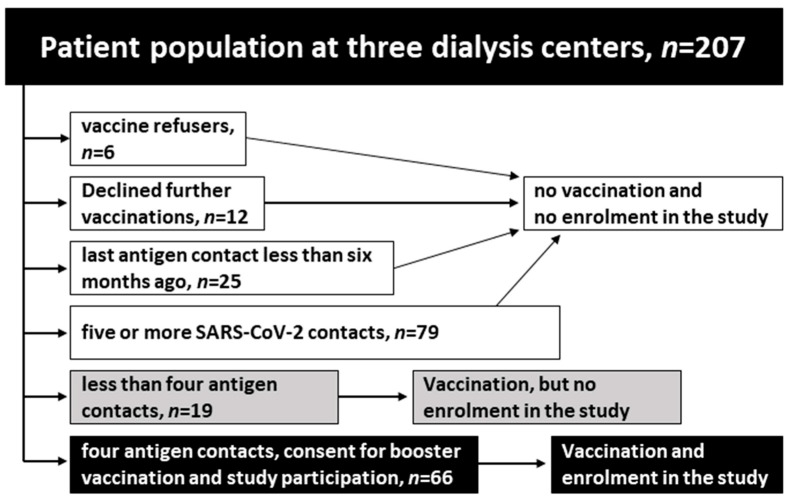
Patient recruitment. Vaccine refusers: no vaccination against COVID-19; declined further vaccinations: received at least a basic immunization (two doses).

**Figure 2 vaccines-11-01428-f002:**
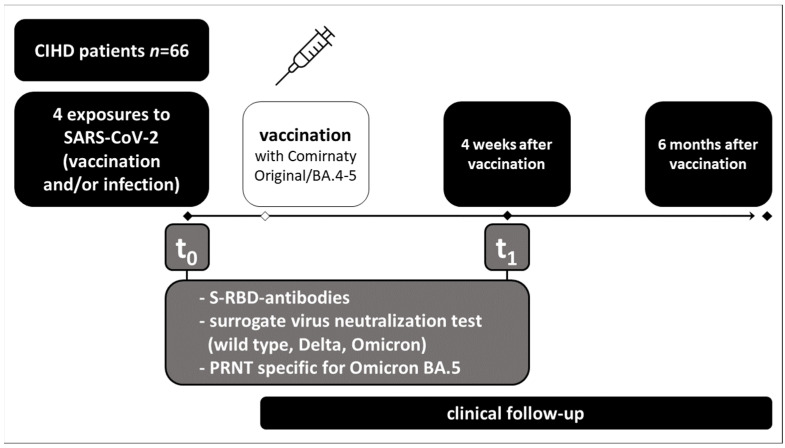
Flowchart of the study setup. CIHD = chronic intermittent hemodialysis, t_0_ = before the vaccination with Comirnaty Original/BA.4-5, six to eight months after the fourth antigen exposure to SARS-CoV-2, and t_1_ = four weeks after the vaccination.

**Figure 3 vaccines-11-01428-f003:**
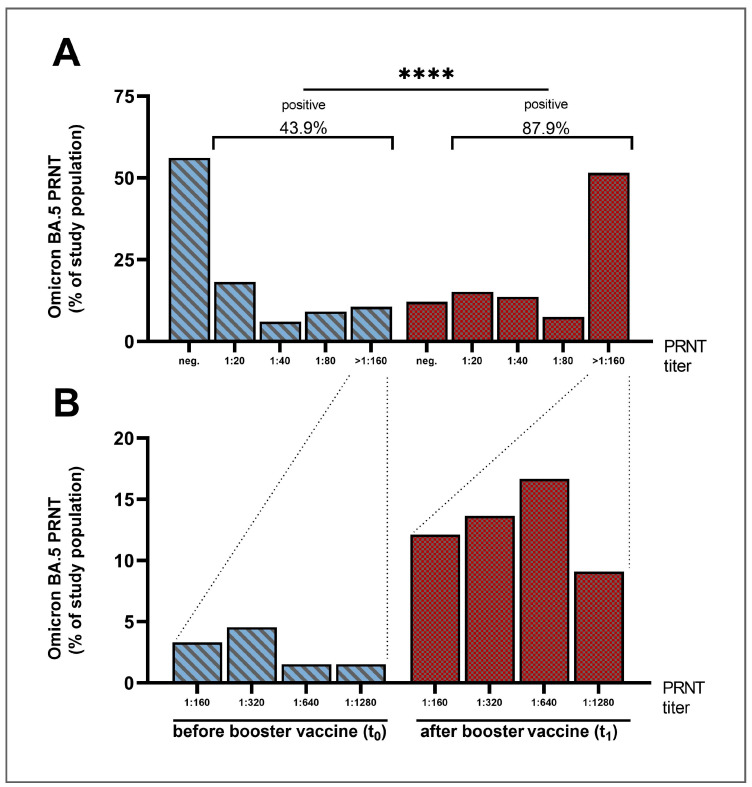
Omicron BA.5-PRNT titers before and four weeks after the vaccination. Percentage of study population grouped by SARS-CoV-2 Omicron BA.5-specific neutralizing antibody titer assessed using PRNT at two time points (t_0_ = before and t_1_ = four weeks after the vaccination with bivalent mRNA vaccine). (**A**) General view. (**B**) Spread in the group with the titers ≥ 1:160. Abbreviations: PRNT, plaque reduction neutralization test, **** (*p* ≤ 0.0001, Friedman test).

**Figure 4 vaccines-11-01428-f004:**
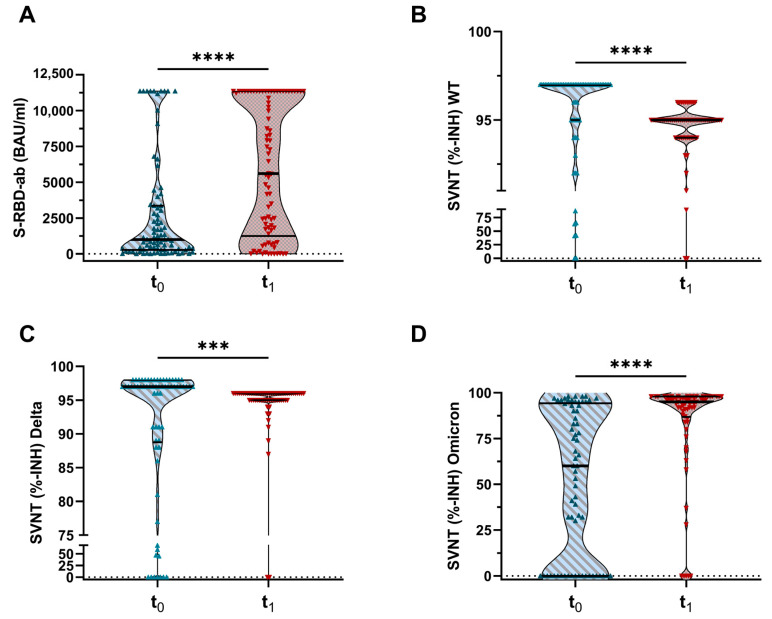
S-RBD-ab and SVNT against wild-type, Delta, and Omicron before and four weeks after the vaccination. Antibody response in plasma samples obtained at two different time points (t_0_ = before the vaccination and t_1_ = four weeks after the vaccination with bivalent mRNA vaccine). (**A**) SARS-CoV-2-specific S-RBD-ab. (**B**) SVNT against wild-type SARS-CoV-2. (**C**) SVNT against Delta variant of SARS-CoV-2 (B.1.617.2). (**D**) SVNT against Omicron variant of SARS-CoV-2 (B.1.1.529). Abbreviations: S-RBD-ab, spike receptor binding domain antibodies; SVNT, surrogate virus neutralization test, WT, wild-type SARS-CoV-2 type (Wuhan Hu-1). Omicron data are expressed as median (IQR). *** (*p* ≤ 0.001), **** (*p* ≤ 0.0001, Friedman test).

**Table 1 vaccines-11-01428-t001:** Basic patient characteristics.

Parameters		Units	All Patients, *n* = 66	
Age (years)		med (IQR)	68	(59.3–80.8)
Gender (male)		*n* (%)	39	(59.1)
Vintage on dialysis (years)		med (IQR)	3.9	(2.8–7.9)
**Ethnicity**				
	Caucasian	*n* (%)	54	(81.8)
	Black	*n* (%)	2	(3)
	Other	*n* (%)	10	(15.2)
**Causes of Kidney Disease**				
	Diabetic nephropathy	*n* (%)	18	(27.3)
	Hypertensive nephropathy	*n* (%)	15	(22.7)
	ADPKD	*n* (%)	5	(7.6)
	Glomerulonephritis	*n* (%)	14	(21.2)
	Chronic interstitial nephritis	*n* (%)	3	(4.5)
	Others/unknown	*n* (%)	11	(16.7)
**Comorbidities**				
Diabetes mellitus		*n* (%)	29	(43.9)
Lung disease		*n* (%)	9	(13.6)
Malignoma		*n* (%)	15	(22.7)
Severe cardiovascular disease		*n* (%)	30	(45.5)
Arterial hypertension		*n* (%)	52	(78.8)
Kidney or liver transplant		*n* (%)	7	(10.6)
Immunosuppression		*n* (%)	7	(10.6)

Abbreviation: ADPKD, autosomal dominant polycystic kidney disease. Data expressed as number (*n*) and percentage (%) or median (med) and interquartile range (IQR), respectively.

**Table 2 vaccines-11-01428-t002:** Overview of used vaccines and antigen contacts.

Exposures to SARS-CoV-2	BNT162b2 (BioNTech/Pfizer)	mRNA-1273 (Moderna)	Infection with SARS-CoV-2
**1st**	43	19 *	4
**2nd**	47	19 *	0
**3rd**	44	20 *	2
**4th**	11	46 †	9
**5th**	66	0	-

Moderna mRNA-1273 vaccine dose: * 0.5 mL = 100 µg, † 0.25 mL = 50 µg.

**Table 3 vaccines-11-01428-t003:** Differences in humoral immunity parameters and clinical features between the Omicron BA.5-PRNT positive and negative patients after vaccination with bivalent mRNA vaccine.

	Units	BA.5-PRNT Positive, *n* = 58	BA.5-PRNT Negative, *n* = 8	*p*-Value
**tests before vaccination (t_0_)**				
S-RBD-ab before vaccination	BAU/mL, med (IQR)	1588 (608–4373)	133 (28–246)	**<0.0001**
wild-type SVNT before vaccination	%-INH, med (IQR)	97.0 (96.0–97.0)	52.5 (30.8–72.5)	**<0.0001**
Delta SVNT before vaccination	%-INH, med (IQR)	97.0 (92.3–98.0)	23.5 (0–66.5)	**<0.001**
Omicron-SVNT before vaccination	%-INH, med (IQR)	71.0 (32.0–95.0)	0 (0)	**<0.001**
Omicron BA.5-PRNT before vaccination	pos. titer (≥1:20), *n* (%)	29 (50%)	0 (0)	**0.0075**
**tests 4 weeks after vaccination (t_1_)**				
S-RBD four weeks after vaccination	BAU/mL, med (IQR)	9701 (5376.8–11,360)	1266 (741–1829)	**<0.0001**
wild-type SVNT four weeks after vaccination	%-INH, med (IQR)	95 (95–95)	94 (70.5–95)	**0.0295**
Delta SVNT four weeks after vaccination	%-INH, med (IQR)	96 (95–96)	95 (70.5–95)	**0.0124**
Omicron-SVNT four weeks after vaccination	%-INH, med (IQR)	96 (92–98)	14 (0–43.5)	**<0.0001**
**clinical parameters**				
Age	years, med (IQR)	68 (59.3–79.8)	67 (59.5–83.3)	0.6783
Body mass index	kg/m^2^, med (IQR)	25.4 (22.7–29.7)	29.2 (26.3–32.7)	0.1456
Lung diseases	% (*n*/total)	13.8 (8/58)	12.5 (1/8)	0.9999
Diabetes	% (*n*/total)	44.8 (26/58)	37.5 (3/8)	0.9999
Cardiovascular disease	% (*n*/total)	46.6 (27/58)	37.5 (3/8)	0.7189
Malignoma	% (*n*/total)	22.4 (13/58)	25 (2/8)	0.9999
Active smoker	% (*n*/total)	15.5 (9/58)	12.5 (1/8)	0.9999
Glomerulonephritis	% (*n*/total)	19 (11/58)	37.5 (3/8)	0.3513
Vintage at dialysis	years, med (IQR)	4.2 (2.9–8.3)	2.9 (2.4–3.6)	0.1296
Transplantation	% (*n*/total)	0	0	n.s.
Immunosuppression	% (*n*/total)	0	37.5 (3/8)	**0.0012**
SARS-CoV-2 infection prior to vaccination	% (*n*/total)	25.9 (15/58)	0	0.1823

Abbreviations: PRNT, plaque reduction neutralization test; S-RBD-ab, spike receptor binding domain antibodies; SVNT, surrogate virus neutralization test; med, median; IQR, interquartile range; pos., positive; bold *p*-values denote statistical significance at the *p* < 0.05 level.

**Table 4 vaccines-11-01428-t004:** Spearman’s rank correlation between S-RBD-ab level and neutralization titers for wild-type, Delta, and Omicron variants of SARS-CoV-2 in surrogate virus neutralization test and Omicron BA.5-PRNT.

Spearman’s Rank Correlation, rho				
before the bivalent vaccination (t_0_)				
	WT SVNT	Delta-SVNT	Omicron-SVNT	Omicron BA.5-PRNT
S-RBD-ab	**0.62**	**0.56**	**0.81**	**0.63**
*p*-value	**<0.0001**	**<0.0001**	**<0.0001**	**<0.0001**
four weeks after the bivalent vaccination (t_1_)				
	WT SVNT	Delta-SVNT	Omicron-SVNT	Omicron BA.5-PRNT
S-RBD-ab	**0.28**	**0.32**	**0.68**	**0.75**
*p*-value	**0.0212**	**0.009**	**<0.0001**	**<0.0001**

Abbreviations: t_0_ = before the vaccination (six to eight months after the 4th antigen exposure to SARS-CoV-2); t_1_ = four weeks after the vaccination with Comirnaty Original/BA.4-5; S-RBD-ab, spike receptor binding domain antibodies; WT, wild-type SARS-CoV-2; SVNT, surrogate virus neutralization test; PRNT, plaque reduction neutralization test; bold rho-values denote strong correlation (>0.50); bold *p*-values indicate statistical significance at the *p* < 0.05 level.

**Table 5 vaccines-11-01428-t005:** Spearman’s rank correlation between neutralization titers for Omicron variant of SARS-CoV-2 in surrogate virus neutralization test and Omicron BA.5-PRNT.

Spearman’s Rank Correlation, rho		
	before the bivalent vaccination (t_0_)	four weeks after the bivalent vaccination (t_1_)
	Omicron BA.5-PRNT	Omicron BA.5-PRNT
Omicron-SVNT	**0.68**	**0.72**
*p*-value	**<0.001**	**<0.001**

Abbreviations: *t*_0_ = before the vaccination (six to eight months after the 4th antigen exposure to SARS-CoV-2); *t*_1_ = four weeks after the vaccination with Comirnaty Original/BA.4-5; SVNT, surrogate virus neutralization test; PRNT, plaque reduction neutralization test; bold rho-values denote strong correlation (>0.50); bold *p*-values indicate statistical significance at the *p* < 0.05 level.

## Data Availability

Data available on request due to privacy restrictions.

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
