# Peer review of "Riding the Omicron BA.5 Wave: Improved Humoral Response after Vaccination with Bivalent Omicron BA.4-5-Adapted mRNA SARS-CoV-2 Vaccine in Chronic Hemodialysis Patients"

_vaccines, 2023, doi:10.3390/vaccines11091428_

Round 1

Reviewer 1 Report

Dear Colleagues,

It was a great pleasure to read the article submitted for review. Since the beginning of the pandemic, we have tried to protect the population against infection and its effects. The thoughts of medical professionals naturally focused especially on immunocompromised patients. This was very visible in dialysis centers (the need to conduct treatment procedures, often in lock-down conditions) and transplant centers (ensuring the safety of patients starting with immunosuppressive treatment). Although reading the introduction for the first time, I had some doubts about the course of the study, the added flowcharts dispelled them. Your paper is well written, easy to read. The limitations of the study you have raised, are completely understandable. Studies on groups of patients in special health conditions or with rare diseases, usually carry this problem. Do you have any information about the occurrence of adverse effects (especially severe) of the vaccination performed, such information would significantly influence the soudnness of the article. Its absence does not discredit your work in general. Can you add this information if available?

Congratulations, I wish you all the best.

Author Response

Response to Reviewer 1 Comments

Dear Colleagues,

It was a great pleasure to read the article submitted for review. Since the beginning of the pandemic, we have tried to protect the population against infection and its effects. The thoughts of medical professionals naturally focused especially on immunocompromised patients. This was very visible in dialysis centers (the need to conduct treatment procedures, often in lock-down conditions) and transplant centers (ensuring the safety of patients starting with immunosuppressive treatment). Although reading the introduction for the first time, I had some doubts about the course of the study, the added flowcharts dispelled them. Your paper is well written, easy to read. The limitations of the study you have raised, are completely understandable. Studies on groups of patients in special health conditions or with rare diseases, usually carry this problem. Do you have any information about the occurrence of adverse effects (especially severe) of the vaccination performed, such information would significantly influence the soudnness of the article. Its absence does not discredit your work in general. Can you add this information if available?

Congratulations, I wish you all the best.

Response:

Dear Colleague,

Thank you very much for your kind feedback!

We added the Results section (Lines 299-300) as follows:

The bivalent vaccine did not cause any moderate or severe adverse events (Grade 2 – 4 on the FDA toxicity grading scale). Mild reactions to the vaccine were not recorded.

Thank you for reviewing this manuscript. 

 we wish you the best as well,  

Eugen Ovcar, and Kirsten de Groot (MD)  

Offenbach am Main, 22/8/2023  

Reviewer 2 Report

The manuscript entitled ''Ride on the Omicron BA.5 wave: improved humoral response 2 after vaccination with bivalent Omicron BA.4-5 adapted mRNA 3 SARS-CoV-2 vaccine in chronic hemodialysis patients'' represents an intersting piece of work discussing improved humoral response following BA.5 exposure in Dialysis patients. I have few concerns related to the study"

1. Abstract- The abstract is written poorly and needs to rephrase concisely and grammatically.

2. M&M- Section Assessment of anti-SARS-CoV-2-spike specific antibodies- the protocol needs to discuss in detail.

Assessment of SARS-CoV-2 Omicron BA.5-specific neutralizing antibodies with a 113 plaque reduction neutralization test (PRNT)- Please describe CaCo-2 cell culture conditions as these cells are hard to grow and expand.

Why human colon cells were chosen why not human lung cells like A549 or Calu-3 cells?

SARS-CoV-2 infection detection- Please mention PCR target gene or kit used for PCR and detailed protocol

3. The authors performed surrogate virus neutralization test against wild type (Wuhan Hu-1), Delta (B.1.617.2) and Omicron (B.1.1.529) and used Live-virus PRNT neutralization for BA.5. Any reason why live-virus neutralization not performed against Wuhan And Delta variants.

4. More specifically, the authors should performed RBD IgG against different variant ELISA.

5. Did the authors performed S protein specific IgA ELISA or RBD IgA ELISA to drive deep into the humoral response?

6. Has the authors tested BAL or Nasal Swabs for IgA?

7. Did the authors try to evalute the Cellular immune response?

8. Section 3.3. Humoral response four weeks after the vaccination (time point t1)- Needs to be expand

9. Please provide conclusions

We demonstrated that immune evasion of SARS-CoV-2 led to a reduced efficiency of 294 the humoral vaccination response induced by monovalent mRNA vaccines. Vaccination 295 with the bivalent Omicron BA.4-5 adapted mRNA vaccine improved the humoral im- 296 mune response of hemodialysis patients against the variant circulating during the study 297 Vaccines 2023, 11, x FOR PEER REVIEW 11 of 13 period. Annually adapted SARS-CoV-2 vaccinations may be necessary in analogy to in- 298 fluenza vaccinations

If this is the conclusion, Sorry it has been a part of Abstract and Introduction.

Author Response

Response to Reviewer 2 Comments

The manuscript entitled ''Ride on the Omicron BA.5 wave: improved humoral response 2 after vaccination with bivalent Omicron BA.4-5 adapted mRNA 3 SARS-CoV-2 vaccine in chronic hemodialysis patients'' represents an interesting piece of work discussing improved humoral response following BA.5 exposure in Dialysis patients. I have few concerns related to the study"

Dear Colleague, 

Thank you for your review. Please find below the answers to your queries and our corrections: 

  1. Abstract- The abstract is written poorly and needs to rephrase concisely and grammatically.

Response:

The abstract has been rephrased.

  1. M&M- Section Assessment of anti-SARS-CoV-2-spike specific antibodies- the protocol needs to discuss in detail.

Assessment of SARS-CoV-2 Omicron BA.5-specific neutralizing antibodies with a plaque reduction neutralization test (PRNT)- Please describe CaCo-2 cell culture conditions as these cells are hard to grow and expand.

Response:

Briefly, Caco-2 (human colon carcinoma) cells were cultured in Minimum Essential Medium (MEM) supplemented with 10% fetal calf serum (FCS), 100 IU/mL of penicillin and 100 g/mL of streptomycin. All culture reagents were purchased from Sigma (St. Louis, MO, USA). The Caco-2 cells were originally obtained from DSMZ (Braunschweig, Germany, no.: ACC 169), differentiated by serial passaging and selected for high permissiveness to virus infection.

Source: 

https://www.mdpi.com/1422-0067/21/12/4396

Why human colon cells were chosen why not human lung cells like A549 or Calu-3 cells?

Response:

As demonstrated in earlier experiments, CaCo-2 cells demonstrated to be very susceptible to SARS-CoV-2 infection in terms of replication and the generation of cytopathogenic effects. 

Source: https://doi.org/10.1056/NEJMc2001899

SARS-CoV-2 infection detection- Please mention PCR target gene or kit used for PCR and detailed protocol

Response:

The SARS-CoV-2 teste was performed as dual target PCR for ORF1 and E genes on a cobas® 6800/8800 system (ROCHE Diagnostics, Mannheim, Germany)

  1. The authors performed surrogate virus neutralization test against wild type (Wuhan Hu-1), Delta (B.1.617.2) and Omicron (B.1.1.529) and used Live-virus PRNT neutralization for BA.5. Any reason why live-virus neutralization not performed against Wuhan And Delta variants.

Response:

Because of organizational and economic reasons (the PRNT is very labour-intensive and must be performed under BSL-3 conditions) and the fact that BA.5 was the dominant circulating variant when the study was conducted.

  1. More specifically, the authors should performed RBD IgG against different variant ELISA.

Response:

Thanks for the remark. As shown in several other studies, anti-Spike IgG (SARS-CoV-2 wild type) ELISAs are used for orientation, because cell-based neutralization assays are the method of choice for the detection of functional neutralizing antibodies against other variants.

  1. Did the authors performed S protein specific IgA ELISA or RBD IgA ELISA to drive deep into the humoral response?

Response:

IgA is considered a secretory antibody which can primarily be found on mucosal membranes. The focus of our study was the detection of systemic IgG antibodies as their presence correlates with (systemic) protection against severe COVID-19.

  1. Has the authors tested BAL or Nasal Swabs for IgA?

Response:

Determination of secretory IgA was not the focus of our study; therefore, no respiratory specimens were tested for the presence of these antibodies.

  1. Did the authors try to evaluate the Cellular immune response?

Response:

Indeed, we investigated the cellular immune response in a subgroup of our patient cohort following the basic immunisation against SARS-CoV-2. We found that the patients in this subgroup that developed humoral immunity also displayed a cellular immune response. Conversely, those who did not experience a humoral response, did not show a cellular reaction, either. As the test for cellular immunity is also very labour intensive and entails a lot of organizational issues because patients dialyse in at least 3 different shifts, we dropped further investigation of cellular immunity.

  1. Section 3.3. Humoral response four weeks after the vaccination (time point t1)- Needs to be expand

Response:

We have expanded the Section 3.3.

  1. Please provide conclusions

We demonstrated that immune evasion of SARS-CoV-2 led to a reduced efficiency of the humoral vaccination response induced by monovalent mRNA vaccines. Vaccination with the bivalent Omicron BA.4-5 adapted mRNA vaccine improved the humoral immune response of hemodialysis patients against the variant circulating during the study period. Annually adapted SARS-CoV-2 vaccinations may be necessary in analogy to in vaccinations

If this is the conclusion, Sorry it has been a part of Abstract and Introduction.

Response:

We have rephrased the discussion and conclusion.

Thank you for reviewing this manuscript.   

   

Eugen Ovcar, and Kirsten de Groot (MD)    

Offenbach am Main, 22/8/2023   

Reviewer 3 Report

In this manuscript, Ovcar et al., investigated the improved humoral response after vaccination with bivalent Omicron BA.4-5 adapted mRNA SARS-Cov-2 vaccine in chronic hemodialysis patients. The title of the paper is in line with the body of the manuscript. The authors have written a clear and  detailed review and the material is well presented. The references used are suitable and updated material, however the limitation of this study is the small size of the cohort. I believe that the article can be accept pending other revision, even if I have below some observations that I think they can make the paper more complete and accurate:

Please check the manuscript for spelling and grammar mistakes

Line 54:  Its effect in hemodialysis patients has not been reported…Why? please clarify in text

Minor editing of English language required

Author Response

Response to Reviewer 3 Comments

Please check the manuscript for spelling and grammar mistakes

Response: Thank you for the advice, we revised the article with the help of a native speaker colleague.

Line 54:  Its effect in hemodialysis patients has not been reported…Why? please clarify in text

Response: At the time of study initiation, no publications were available on the effect of bivalent vaccination on haemodialysis patients.  In the approval studies and first publications, there was no information regarding chronic kidney failure [s. below].

We added the Introduction (Lines 65-66):

In immunocompetent patients, this led to an improved humoral vaccination response [10, 11]. Its effect in hemodialysis patients has not been reported were not evaluated in the approval studies und first publications.

Thank you for reviewing this manuscript. 

Sincerely,  

Eugen Ovcar, and Kirsten de Groot (MD)  

Offenbach am Main, 22/08/2023  

  1. Davis-Gardner ME, Lai L, Wali B, Samaha H, Solis D, Lee M, Porter-Morrison A, Hentenaar IT, Yamamoto F, Godbole S, Liu Y, Douek DC, Lee FE, Rouphael N, Moreno A, Pinsky BA, Suthar MS. Neutralization against BA.2.75.2, BQ.1.1, and XBB from mRNA Bivalent Booster. N Engl J Med. 2023 Jan 12;388(2):183-185. doi: 10.1056/NEJMc2214293. Epub 2022 Dec 21. PMID: 36546661; PMCID: PMC9812288.

  1. Zou J, Kurhade C, Patel S, Kitchin N, Tompkins K, Cutler M, Cooper D, Yang Q, Cai H, Muik A, Zhang Y, Lee DY, Åžahin U, Anderson AS, Gruber WC, Xie X, Swanson KA, Shi PY. Neutralization of BA.4-BA.5, BA.4.6, BA.2.75.2, BQ.1.1, and XBB.1 with Bivalent Vaccine. N Engl J Med. 2023 Mar 2;388(9):854-857. doi: 10.1056/NEJMc2214916. Epub 2023 Jan 25. PMID: 36734885; PMCID: PMC9891359.

European Medicine Agency: https://www.ema.europa.eu/en/documents/variation-report/comirnaty-h-c-005735-ii-0140-epar-assessment-report-variation_en.pdf

Reviewer 4 Report

This is an important and interesting study that investigated the humoral immune response in 66 hemodialysis patients after receiving a bivalent mRNA Sars-CoV-2 vaccine following pre-exposure by infection or/and vaccination. Despite the increase in antibody levels and their efficacy in biological or biochemical tests, the authors observed breakthrough infection in 11 of the study participants. A comparison is shown for the general population in the geographical area of interest.

Overall, the study is well conducted. One main focus is the surrogate viral neutralization test, which is a highly artificial test procedure and may not adequately reflect the biological mechanism of virus entry into cells. A shortcoming of the study is that no T-cell response is documented.

Minor comment: The criteria for how the cytopathic effects were defined in the plaque reduction neutralization test need to be described in the methods section.

Author Response

Response to Reviewer 4 Comments

It is an important and interesting study that investigated the humoral immune response in 66 hemodialysis patients after receiving a bivalent mRNA Sars-CoV-2 vaccine following pre-exposure by infection or/and vaccination. Despite the increase in antibody levels and their efficacy in biological or biochemical tests, the authors observed breakthrough infection in 11 of the study participants. A comparison is shown for the general population in the geographical area of interest.

Overall, the study is well conducted. One main focus is the surrogate viral neutralization test, which is a highly artificial test procedure and may not adequately reflect the biological mechanism of virus entry into cells. A shortcoming of the study is that no T-cell response is documented.

Minor comment: The criteria for how the cytopathic effects were defined in the plaque reduction neutralization test need to be described in the methods section.

Response:

Details are described earlier – see S. Westhaus and HF Rabenau „Neutralization Assay for SARS-CoV-2 Infection: Plaque Reduction Neutralization Test“ (Methods in molecular biology (Clifton, N.J.) 2452:353-360, 2022, DOI:10.1007/978-1-0716-2111-0_20, In book: SARS-CoV-2 (pp.353-360)

We added the text (Lines 154-155)

Thereafter, CaCo-2 cells were analyzed for cytopathic effect (CPE) formation by light microscopy to define the neutralization titers as reciprocal value of the highest dilution of serum that prevented infection of 50% of the cells. The criteria for CPE are described in detail before [16]. Each serum sample was tested in duplicate, in the case of discrepancies, the lowest observed titer was chosen. Cut-off for positivity was set to ≥1:20 dilution.

Thank you for reviewing this manuscript. 

Sincerely,  

Eugen Ovcar, and Kirsten de Groot (MD)  

Offenbach am Main, 22/08/2023  

Round 2

Reviewer 2 Report

Thanks for drafting the changes in the manuscript and response to the questions. The manuscript has no issues from my side.